# Effects of Caprylic Acid and Eicosapentaenoic Acid on Lipids, Inflammatory Levels, and the JAK2/STAT3 Pathway in ABCA1-Deficient Mice and ABCA1 Knock-Down RAW264.7 Cells

**DOI:** 10.3390/nu15051296

**Published:** 2023-03-06

**Authors:** Xinsheng Zhang, Peng Zhang, Yinghua Liu, Zhao Liu, Qing Xu, Yong Zhang, Lu Liu, Xueyan Yang, Liya Li, Changyong Xue

**Affiliations:** 1Department of Nutrition, The First Medical Center, Chinese PLA General Hospital, Beijing 100853, China; 2Guizhou Crops of Chinese People’s Armed Police Force, Guiyang 550001, China

**Keywords:** caprylic acid, EPA, inflammatory cytokine, ABCA1, JAK2, STAT3

## Abstract

Our previous studies have found that caprylic acid (C8:0) can improve blood lipids and reduce inflammation levels and may be related to the upregulation of the p-JAK2/p-STAT3 pathway by ABCA1. This study aims to investigate the effects of C8:0 and eicosapentaenoic acid (EPA) on lipids, inflammatory levels, and the JAK2/STAT3 pathway in ABCA1-deficient mice (ABCA1^−/−^) and ABCA1 knock-down (ABCA1-KD) RAW 264.7 cells. Twenty 6-week ABCA1^−/−^ mice were randomly divided into four groups and fed a high-fat diet, or a diet of 2% C8:0, 2% palmitic acid (C16:0) or 2% EPA for 8 weeks, respectively. The RAW 264.7 cells were divided into the control or control + LPS group, and the ABCA1-KD RAW 264.7 cells were divided into ABCA1-KD with LPS (LPS group), ABCA1-KD with LPS + C8:0 (C8:0 group), and ABCA1-KD with LPS + EPA (EPA group). Serum lipid profiles and inflammatory levels were measured, and ABCA1 and JAK2/STAT3 mRNA and protein expressions were determined by RT-PCR and Western blot analyses, respectively. Our results showed that serum lipid and inflammatory levels increased in ABCA1^−/−^ mice (*p* < 0.05). After the intervention of different fatty acids in ABCA1^−/−^ mice, TG and TNF-α were significantly lower, while MCP-1 increased significantly in the C8:0 group (*p* < 0.05); however, LDL-C, TC, TNF-α, IL-6, and MCP-1 levels decreased significantly and IL-10 increased significantly in the EPA group (*p* < 0.05). In the aorta of ABCA1^−/−^ mice, C8:0 significantly decreased p-STAT3 and p-JAK2 mRNA, while EPA significantly reduced TLR4 and NF-κBp65 mRNA. In the ABCA1-KD RAW 264.7 cells, TNF-α and MCP-1 were increased significantly and IL-10 and IL-1β were significantly decreased in the C8:0 group (*p* < 0.05). The protein expressions of ABCA1 and p-JAK2 were significantly higher, and the NF-κBp65 was significantly lower in the C8:0 and EPA groups (*p* < 0.05). Meanwhile, compared to the C8:0 group, the NF-κBp65 protein expression was significantly lower in the EPA group (*p* < 0.05). Our study showed that EPA had better effects than C8:0 on inhibiting inflammation and improving blood lipids in the absence of ABCA1. C8:0 may be involved mainly in inhibiting inflammation through upregulation of the ABCA1 and p-JAK2/p-STAT3 pathways, while EPA may be involved mainly in inhibiting inflammation through the TLR4/NF-κBp65 signaling pathway. The upregulation of the ABCA1 expression pathway by functional nutrients may provide research targets for the prevention and treatment of atherosclerosis.

## 1. Introduction

Atherosclerotic cardiovascular disease (ASCVD) is one of the leading causes of mortality worldwide [1]. In particular, in low- and middle-income countries, ASCVD accounts for about 80% of the disease burden [2]. Atherosclerosis (AS) is the pathological basis of ASCVD, and is closely related to various risk factors, including hyperlipidemia, hypertension, chronic inflammation, immune factors, etc. Exploring the target of inhibiting AS has always been a hot topic in the treatment of ASCVD.

ATP-binding box transporter A1 (ABCA1) is a membrane protein that can promote intracellular cholesterol efflux and promote liver HDL production. Current studies have shown that ABCA1 not only promotes cholesterol reversal and reduces AS lipid deposition, but also participates in the inflammatory reaction process of AS. ABCA1 has been reported to inhibit the inflammatory response in the following ways [3,4,5]: first, ABCA1 directly activates Janus kinase 2 (JAK2) and the signal transducer and activator of transcription 3 (STAT3) downregulating downstream signaling molecules; the second is that ABCA1 indirectly inhibits the toll-like receptor-4 (TLR4) signaling pathway by promoting cholesterol efflux and reducing lipid raft in the cell membrane. These findings suggest that ABCA1 can play a direct role in cardioprotective effects by promoting cholesterol transport and inhibiting inflammation.

Dietary fatty acids are considered important factors affecting the progress of AS. Studies have reported that saturated fatty acids (SFAs) can increase the level of low-density lipoprotein cholesterol (LDL-C), thus promoting the occurrence of AS, while unsaturated fatty acids can improve blood lipids and play a protective role in cardiovascular diseases, especially omega-3 polyunsaturated fatty acids (ω-3 PUFAs), such as eicosapentaenoic acid (EPA) and docosahexaenoic acid (DHA) [6]. Fatty acids and inflammatory factors are closely related to the progress of AS [7]. Current studies suggest that PUFAs, such as linoleic acid (C18:2), omega-6 polyunsaturated fatty acids (ω-6 PUFAs), and ω-3 PUFAs from food sources, can reduce inflammation. Fish oil is rich in ω-3 PUFAs (DHA and EPA) and has anti-inflammatory effects. Its anti-inflammatory mechanism has been reported to be through inhibition of the TLR4 signaling pathway [8]. Medium-chain fatty acids (MCFAs), including caprylic acid (C8:0) and capric acid (C10:0), occur in milk fat, palm oil, and various feed materials [9]. MCFAs differ from long chain fatty acids (LCFAs) in digestion, absorption, and metabolism, and studies have found that MCFAs can reduce body weight and improve internal fat accumulation [10,11] and cholesterol metabolism [12,13]. Our previous experiments have confirmed that MCFAs can upregulate ABCA1 gene and protein expression in the apoE-deficient mouse liver, and C8:0 can promote cholesterol efflux in macrophages [14]. Further studies found that C8:0 could inhibit the levels of inflammatory cytokines in RAW 264.7 cells and in the serum of apoE-deficient mice, and confirmed that C8:0 could inhibit the inflammatory response based on the ABCA1-mediated JAK2/STAT3 pathway [15].

JAK2 is activated by ABCA1, undergoes autophosphorylation, and then phosphorylates its downstream target STAT3 [16], and then regulates the levels of nuclear factor kappa Bp65 (NF-κBp65), tumor necrosis factor α (TNF-α), interleukin-6 (IL-6), and monocyte chemoattractant protein-1 (MCP-1), which leads to the occurrence of AS [17]. This study aims to investigate the effects of C8:0 and EPA on lipid and inflammatory levels, and the JAK2/STAT3 pathway in ABCA1-deficient mice (ABCA1^−/−^) and ABCA1 knock-down (ABCA1-KD) RAW 264.7 cells, and to confirm that C8:0 plays a regulatory role in improving blood lipid and inflammation mainly through ABCA1.

## 2. Materials and Methods

### 2.1. Materials

Fetal bovine serum (FBS), lipopolysaccharide (LPS), DMEM culture medium, and bovine serum albumin (BSA) were provided by Gibco (Grand Island, NE, USA). C8:0, palmitic acid (C16:0), and EPA were obtained from Sigma-Aldrich (St. Louis, MO, USA). Other reagents were available from Sigma-Aldrich.

### 2.2. Feed Configuration

For feed content, we applied the same method as our previous research [18]. Briefly, based on high-fat feed, 2% of the same amounts of different kinds of fatty acids were added. Intervention feeds included a high-fat diet (HFD group), and high-fat diets of 2% C8:0 (C8:0 group), 2% eicosapentaenoic acid (EPA group), and 2% palmitic acid (C16:0 group). The feed was obtained from Beijing Huafukang Biotechnology Co., Ltd. (license No.: SCXK 2014-0008). The ingredient list and fatty acid compositions of the intervention diets are provided in Appendix A.

### 2.3. Experimental Animals

To obtain ABCA1 knockout mice, we used DBA/1-ABCA1 tm1jdm/J female mice (ABCA1 heterozygous mice) purchased from Jackson Laboratory (stock#003897, Bar Harbor, ME). Because ABCA1 homozygotes cannot reproduce and have a 50% mortality rate, we used male C57BL/6J mice and heterozygous ABCA1 mice to reproduce more heterozygous mice and then obtained 20 homozygous female and male ABCA1 mice at 5 weeks of age. All animals were genetically identified prior to use to confirm that they were ABCA1^−/−^ mice. Five animals per cage were housed in polycarbonate cages; temperature was maintained at 21–23 °C and humidity was maintained at 40–60%, with a 12 h light/dark cycle. Both the ABCA1^−/−^ mice and the same week-old C57BL/6J mice were kept on a HFD prior to intervention, and the HFD was replaced with intervention diets for 8 weeks after random allocation based on fasting weight. Fasting weight was measured weekly during this period (fasting did not limit drinking water at night prior to measurement). The bedding and drinking water of the mice were replaced every 2–3 days and the feed intake of the mice was recorded. All experimental procedures were approved by the Animal Care and Use Committee of the Chinese PLA General Hospital.

### 2.4. Preparation of Fatty Acids

Fatty acids were prepared as in our previous research [15]. Briefly, the fatty acids were dissolved in a 95% ethanol solution and then diluted with serum-free medium containing 20 mg/mL of BSA, with 100 mmol/L added to the culture hole and 50 ng/mL of LPS added to the culture wells. Before the cell experiment, the obtained solution was incubated at 37 °C for 1 h.

### 2.5. ABCA1-KD in RAW 264.7 Cells

The RAW 264.7 cell line was obtained from the Peking Union Medical College, and the cells were cultured in DMEM with heat-inactivated FBS (10%) and penicillin–streptomycin solution (1%) in a humidified incubator with 95% air and 5% CO_2_ at 37 °C. RAW 264.7 cells at the logarithmic growth stage were inoculated into 6-well cell culture plates (2 × 10^5^ cells per well) and cultured overnight in an incubator. The ABCA1-KD RAW 264.7 cells were constructed with three types of siRNA of ABCA1-1530, ABCA1-1701, and ABCA1-4931 (Appendix A). The most effective plasmids that inhibited ABCA1 were screened by RT-PCR, as shown in Figure 1. ABCA1-1701 had the strongest inhibitory level and was selected to construct a siRNA plasmid to obtain ABCA1-KD RAW 264.7 cells. Two hours before transfection, the culture medium was changed to serum-free DMEM. According to the plasmid transfection instructions, cultured RAW 264.7 cells (2 × 10^5^ cells/well) were transfected with the ABCA1-1701 plasmid using Lipofectamine 2000 reagent (Life Technologies, Carlsbad, CA, USA). The cells were then tested with the optimal concentration of G418 (500 μg/mL) for about 3 weeks. A limited dilution method was applied to isolate and obtain the maximum number of stably transfected cells. DMEM medium containing 10% fetal bovine serum was used during the experiment and then refilled with LPS medium (final concentration 100 ng/mL), and incubated for another 24 h after the addition of C8:0 or EPA to the culture medium. The ABCA1-KD RAW 264.7 cells were randomly divided into 5 groups (n = 5), including the control group (RAW 264.7 cells), ABCA1-KD group, ABCA1-KD + LPS group (LPS 100 ng/mL), ABCA1-KD + LPS + C8:0 (LPS 100 ng/mL, C8:0 100 μmol/L), and ABCA1-KD + LPS + EPA (LPS 100 ng/mL, EPA 100 μmol/L). Then the levels of interleukin-1β (IL-1β), IL-6, interleukin-10 (IL-10), TNF-α, and MCP-1 in the cell lysate were detected according to the instructions of the ELISA kit. Cell assay was repeated, and the protein expressions of ABCA1, JAK2/STAT3, p-JAK2/p-ATAT3, and NF-κBp65 were determined by Western blot analyses.

### 2.6. Serum Lipid Profiles Measurement

Serum triglyceride (TG), total cholesterol (TC), high-density lipoprotein cholesterol (HDL-C), and LDL-C (Abcam, Cambridge, UK) were determined according to the commercial kit instructions, and HDL-C/LDL-C was calculated.

### 2.7. Inflammatory Level Measurement

After 8 weeks, the mice were sacrificed by intramuscular injection of 10 mg/kg of xylazine hydrochloride, blood was drawn from the abdominal aorta and then centrifuged at 4 °C and 3000 r/min for 10 min, and serum was collected for detection. After the cell experiment, the cell lysate from each group was collected and centrifuged at 3000 r/min at 4 °C for 10 min, and the supernatant was collected to be measured. The IL-1β, IL-6, IL-10, TNF-α, and MCP-1 were determined following the instructions of the ELISA kit (R&D Systems, Minneapolis, MN, USA).

### 2.8. Real-Time PCR Analysis

For RNA expression analysis, about 50 mg of aorta samples were taken, total RNA was isolated using TRIzol reagent (Omega Bio-Tek, Norcross, GA, USA), and then reverse transcription was performed using a reverse transcription kit (NEB, M-MLV kit). The reaction mixtures were incubated at 95 °C for 2 min for the initial denaturation, followed by 45 cycles of 25 °C/5 min, 50 °C/15 min, 85 °C/5 min, and 4 °C/10 min for cDNA, and then 50 °C/2 min, 95 °C/10 min, 95 °C/30 s, and 60 °C/30 s. Relative expression levels were calculated with the ΔCt method. Primers were designed using Primer Express Software v3.0 (Applied Biosystems, SAN Jose, California, USA) (Table 1).

### 2.9. Western Blot Analysis

The 20 mg of mouse aorta tissue sample was extracted by protein lysis buffer and the cells were extracted with RIPA buffer (CST). Western blot analysis of mouse aorta tissue and cell samples referred to previous studies [18]. Immunoblotting for STAT3 (abcam, no.ab68153, 1:1000), JAK2 (abcam, no.ab108596, 1:1000), p-STAT3 (abcam, no.ab76315, 1:800), p-JAK2 (abcam, no.ab32101, 1:800), ABCA1 (abcam, no.Ab18180, 1:200), NF-κBp65 (abcam, no.ab32536, 1:1000), MYD88 (abcam, no.ab219413, 1:1000), TLR4 (Proteintech, no.19811-1-AP, 1:1000), and β-actin (Proteintech, no.66009-1-Ig, 1:5000) followed the procedures. The bands were visualized using a chemiluminescence detection system.

### 2.10. Statistical Analysis

Based on our preliminary experiment [15], the sample size was estimated using G*Power software v3.1.9.3 (Heinrich-Heine University, Germany). With power = 80%, α = 0.05, effect size = 0.85, the minimum sample size should be 5. In mouse experiments, the sample sizes for analysis of inflammatory levels, blood lipids, PCR, and Western blot were 5, 5, 5, and 4, respectively. The sample sizes for the analysis of inflammatory cytokines and Western blotting in cell experiments was 5 and 3, respectively. All data are expressed as mean ± standard deviation and for the detection of a significant difference (*p* < 0.05, two-tailed). The normality of the data was analyzed by Shapiro–Wilk test. The normal distribution data between the two groups were analyzed by Student’s t test, and the non-normal distribution data between the two groups were analyzed by Mann–Whitney U and Wilcoxon signed-rank tests. One-way analysis of variance was used to analyze the multigroup data, and Tukey–Kramer multiple comparison analysis was used to analyze the differences between groups. SPSS 28.0 (SPSS, Inc., Chicago, IL, USA) was used to analyze the research data.

## 3. Results

### 3.1. Body Weight of ABCA1^−/−^ Mice

After 2 weeks of intervention, the fasting body weight of the EPA, C8:0, and C16:0 groups decreased significantly compared to that of the HFD group (*p* < 0.05) (Figure 2A). There were no significant differences in the average feed intake among all groups during the intervention period (*p* > 0.05) (Figure 2B).

### 3.2. Serum Lipid Profiles in ABCA1^−/−^ Mice

After 8 weeks of intervention, ABCA1^−/−^-HFD mice showed a marked reduction in TC, TG, and non-HDL-C (*p* < 0.05) (Figure 3A,B,F). Next, we analyzed the serum lipids of ABCA1^−/−^ mice with different fatty acid HFD. The EPA group had a significantly lower level of TC than that of the HFD, C8:0, and C16:0 groups (*p* < 0.05) (Figure 3A). The C8:0 and C16:0 groups had a significantly lower TG level than the HFD and EPA groups (Figure 3B) and had a significantly higher HDL-C level than the EPA group (*p* < 0.05) (Figure 3C). The EPA group exhibited a significant decrease in serum LDL-C and non-HDL-C levels compared to those of the C8:0 and C16:0 groups (*p* < 0.05) (Figure 3D,F).

### 3.3. Serum Inflammatory Factors in ABCA1^−/−^ Mice

ABCA1 knockout can promote the release of different pro-inflammatory cytokines in mice. According to Figure 4, ABCA1^−/−^-HFD mice had significantly higher levels of IL-1β, IL-6, TNF-α, and MCP-1, and had a significantly lower level of IL-10 than those of WT-HFD mice (*p* < 0.05). Then we analyzed the effects of different fatty acid HFDs on serum inflammation in ABCA1^−/−^ mice. The EPA group exhibited a significant decrease in serum IL-1β, IL-6, TNF-α, and MCP-1 and a significant increase in serum IL-10 compared to that of the C8:0, C16:0, and HFD groups (*p* < 0.05) (Figure 4). Although the C8: 0 group had a significantly lower level of TNF-α than that of the HFD and C16: 0 groups, it had a significantly higher level of MCP-1 than that of the HFD group (*p* < 0.05) (Figure 4A,D).

### 3.4. The mRNA Expression Levels of TLR4 and JAK2/STAT3 in the ABCA1^−/−^ Mouse Aorta

In ABCA1^−/−^ mice, compared to the HFD group, the mRNA expression of TLR4 was significantly downregulated in the C8:0, EPA, and C16:0 groups (*p* < 0.05) (Figure 5D). EPA group mice had significantly lower TLR4 and NF-κBp65 mRNA expressions than those of the HFD and C16:0 groups, and had a significantly lower TLR4 mRNA expression than that of the C8:0 group (*p* < 0.05) (Figure 5D,F).

### 3.5. The Relative Protein Expression Levels of TLR4 and JAK2/STAT3 in the ABCA1^−/−^ Mouse Aorta

In ABCA1^−/−^ mice, the C8:0 group had significantly lower expression levels of p-STAT3 and p-JAK2 than those of the HFD group (*p* < 0.05) (Figure 6B,C). The EPA group had a significantly lower expression level of NF-κBp65 than that of the HFD and C16:0 groups, and had a significantly lower expression level of TLR4 than that of the HFD, C16:0, and C8:0 groups (*p* < 0.05) (Figure 6D,F).

### 3.6. The Inflammatory Factors of ABCA1-KD RAW 264.7 Cells

After RAW 264.7 cells induced by LPS, the levels of TNF-α, MCP-1, IL-6, and IL-1β were significantly increased, while the levels of IL-10 decreased significantly (*p* < 0.05) (Figure 7A–E). Similarly, the levels of TNF-α, MCP-1, IL-6, and IL-1β inflammatory cytokines in the ABCA1-KD + LPS group decreased significantly compared to the control group + LPS group, while the level of IL-10 increased significantly (*p* < 0.05) (Figure 7A–E). In ABCA1-KD RAW 264.7 cells with LSP, the EPA group exhibited a significant decrease in TNF-α, MCP-1, IL-6, and IL-1β and a significant increase in IL-10 compared to that of the LPS groups (*p* < 0.05) (Figure 7). In addition, the EPA group had a significantly lower level of TNF-α and MCP-1 than that of the C8: 0 group (Figure 7A,B); however, it had significantly higher levels of IL-1β and IL-10 than those of the C8: 0 group (*p* < 0.05) (Figure 7D,E). The C8:0 group had a significantly higher level of TNF-α than that of the LPS group (Figure 7A), while it had significantly lower levels of MCP-1, IL-6, IL-1β, and IL-10 than those of the LPS group (*p* < 0.05) (Figure 7B–E).

### 3.7. The Protein Expression of JAK2/STAT3 in ABCA1-KD RAW 264.7 Cells

After LPS-induced RAW 264.7 cells, the ABCA1 expression was significantly lower, whereas the NF-κBp65, p-JAK2, and p-STAT3 expressions were significantly higher (*p* < 0.05) (Figure 8D,F,H). Additionally, after LPS-induced ABCA1-KD RAW 264.7 cells, the ABCA1, p-STAT3, and p-JAK2 expressions were significantly lower than those of the Control + LPS group, whereas the NF-κBp65 expression was significantly higher than that of the control + LPS group (*p* < 0.05) (Figure 8B,D,F,H). The C8:0 group had significantly higher expressions of ABCA1 and p-JAK2 than those of the ABCA1-KD + LPS group, while they had significantly lower expression of NF-κBp65 than that of the ABCA1-KD + LPS group (*p* < 0.05) (Figure 8B,D,F). Similarly, the EPA group had significantly higher expressions of ABCA1 and p-JAK2 than those in the ABCA1-KD + LPS group (*p* < 0.05) (Figure 8B,D,F), but had a significantly lower expression of NF-κBp65 than that of the C8:0 group (*p* < 0.05) (Figure 8F).

## 4. Discussion

The present study showed that ABCA1 knockout resulted in dyslipidemia and increased inflammation in mice, which also resulted in significant fasting weight loss in the C8:0, C16:0, and EPA groups. Aiello et al. [19] reported that the survival rate and weight gain of ABCA1^−/−^ mice after weaning were similar to those of wild-type mice, and the weight range of ABCA1^−/−^ mice in this study was basically the same. In addition to ABCA1 defects, dietary differences have also been suggested as possible causes of weight loss. There was no significant difference in body weight between HFD-fed ABCA1^−/−^ and wild-type mice in this study. On the contrary, Orso et al. [20] reported that ABCA1 knockout may cause insufficient vitamin absorption and platelet aggregation, as well as severe small intestinal lesions, resulting in decreased survival and body weight. In addition, homozygous female ABCA1-deficient mice are difficult to breed, probably due to altered hormone secretion and subsequent placental abnormalities caused by reduced estrogen and progesterone levels [21], which may also affect their metabolism and development. More recently, the important beneficial role that ABCA1 plays in modulating inflammation has been realized [22]. In ABCA1^−/−^ mice, we found that C8:0 did not significantly improve LDL-C, TC, and HDL-C/LDL-C except for reducing TG, while EPA significantly improved LDL-C and TC, and a consistent effect was also observed on inflammation in ABCA1^−/−^ mice and ABCA1-KD RAW 264.7 cells. Furthermore, C8:0 group mice had significantly decreased expression of p-STAT3 and p-JAK2 in the aorta, while EPA significantly decreased the expression of TLR4 and NF-κBp65 in the aorta of ABCA1^−/−^ mice and ABCA1-KD RAW 264.7 cells. These results differ from our previous study of C57BL/6J mice [15]. These findings may help explore the different mechanisms of C8:0 and EPA in the regulation of blood lipids and inflammation.

ABCA1 belongs to the ABCA subfamily. ABCA1 was found to be highly expressed in hepatocytes, intestinal cells, macrophages, and endothelial cells [23]. Studies have shown that ABCA1 plays a crucial role in cholesterol reversal [22]. Fatty acids have been reported to regulate ABCA1 expression in mouse models by activating liver cyclic AMP-dependent protein kinase A and LXR/RXR pathways [4]. For example, linoleic acid suppressed the levels of ABCA1 transcripts and protein in human macrophages [24]. On the contrary, palmitic acid, ω-6 PUFAs and linolenic acid as a precursor to EPA, had the opposite effect [24,25]. In our previous studies, we found that MCT reduced LDL-C and TC levels and improved HDL-C levels in patients with high triglycerides [26,27]. We also observed that C8:0 could reduce TC and LDL-C levels, increase the HDL-C/LDL-C ratio, and improve atherosclerosis in apoE-deficient mice [18]. In recent years, in our mouse experiments, C8:0 was found to upregulate the expression of ABCA1 in the liver [14], in the mouse aorta [15], and in RAW 264.7 cells [15]. Tangier’s disease is a high-risk ASCVD disease due to the lack of ABCA1, leading to high TG and TC and low HDL [28]. In this study, we found that ABCA1 knockout increased TG, TC, and non-HDL-C, but HDL-C did not decrease significantly, which may be related to the different intervention feeds and compensatory mechanisms. Drobnik et al. fed ABCA1^+/+^ and ABCA1^−/−^ mice with a cholesterol-free diet for 14 days and found a significant decrease in both serum HDL-C and TC in ABCA1^−/−^ mice [29]. Haghpassand et al. reported that high fat feeding increased HDL cholesterol and apoA1 levels in wild-type mice or bone marrow-transplanted ABCA^−/−^ mice [30]. A single deficiency of ABCA1 or ABCG1 in macrophages has been reported to not increase atherosclerosis, probably because ABCA1 deficiency leads to upregulation of ABCG1 expression [31]. Similarly, ABCG1-deficient mice were shown to have decreased plasma HDL cholesterol levels when fed a high-cholesterol diet. In addition to the significant reduction in TG in C8:0, there was no significant improvement in LDL-C, TC, and non-HDL-C levels. In contrast, EPA significantly reduced LDL-C, TC, and non-HDL-C levels compared to C8:0. It is suggested that the mechanism of C8:0 and EPA in reducing lipids is different, which is worthy of further study.

Research evidence has supported the role of ABCA1 in the regulation of cholesterol efflux [32] and its anti-inflammatory effects [33]. ABCA1 can regulate inflammation by participating in cellular cholesterol and phospholipid transport and the formation of lipid domains on the cell surface [3,33,34]. Dietary fatty acids not only affect blood lipids but also mediate inflammation levels, such as the way in which excessive intake of SFAs can increase the level of serum inflammatory cytokines in animals [35]. Furthermore, palmitic acid and stearic acid promoted the expression of TNF-α and IL-1β in macrophages [36]. Diets rich in fish oil can downregulate the expression of TLR4, TNF-α, IL-1, nucleotide-binding oligomerization domain protein1, and nucleotide-binding oligomerization domain protein2 in the liver of piglets [8]. Furthermore, supplementation with highly purified concentrated fish oil increased the levels of IL-10, IL-12, and IFN-γ while decreasing the levels of TNF-α and IL-6 [37]. In addition, EPA and DHA pretreatment may be beneficial for vascular inflammation in human saphenous veins undergoing a coronary bypass operation [38]. Although C8:0 belongs to SFA, both C8:0 and EPA can decrease the levels of MCP-1 and TNF-α and increase the level of IL-10 in mice and cells treated with LPS, in line with previous findings [15,18]. In mice, ABCA1 knockout increases inflammatory infiltration in vascular walls, peritoneal cavities, and blood circulation [19]. ABCA1/G1 deficiency improved LPS-induced inflammatory gene expression in mouse aortic endothelial cells [31]. In addition, THP-1 macrophage knockdown of ABCA1 inhibits downregulation of inflammatory cytokines by the apolipoprotein A-1 binding protein [39]. Patients with ABCA1 dysfunction tend to have chronic inflammation, suggesting that ABCA1 has a regulatory role in inflammation [40]. In this study, we found a significant decrease in inflammation levels by EPA after ABCA1 knockout; however, only TNF-α levels were significantly reduced by C8:0. In the ABCA1-KD RAW 264.7 cell assay, EPA was also found to significantly reduce inflammation levels compared to C8:0. The anti-inflammatory effects of C8:0 and EPA were different in ABCA1 deficiency. ABCA1 regulates both lipid metabolism and inflammation and may be the key protein in the mechanism of action of C8:0 and ω-3 PUFAs.

ABCA1 suppresses inflammation through multiple mechanisms. ABCA1 regulates the inflammatory response through NF-κBp65, TLR4/MYD88, JAK2/STAT3, cAMP/PKA, and apoptosis pathways [4,41]. SFAs, especially lauric acid, palmitic acid, and stearic acid, have been found to increase the level of IL-6 expression in macrophages through the TLR4 pathway, and stearic acid can promote the release of MCP-1 by activating TLR4 [42]. The mechanism of ω-6 PUFAs inhibiting the inflammatory response includes inhibition of the TLR-4/MYD88/NF-κBp65 pathway [43] and activation of GPR120 to inhibit the TAK1/NF-κBp65/JNK pathway [44]. Our previous results suggest that C8:0 can inhibit inflammation and improve atherosclerosis through the TLR4/NF-κBp65 signaling pathway in apoE^−/−^ mice [18]. Further studies showed that C8:0 plays an important role in both lipid metabolism and inflammation, which may be related to the signaling pathways ABCA1 and JAK2/STAT3 [15]. Compared with EPA, the transcription levels of ABCA1, JAK2, and STAT3 in the mouse aorta increased significantly in C8:0, but there was no significant difference in the expressions of JAK2 and STAT3 in LPS-stimulated RAW 264.7 cells [15]. In this study, C8:0 significantly reduced the p-STAT3 and p-JAK2 in the aorta of ABCA1^−/−^ mice. However, EPA significantly inhibited TLR4 and NF-κBp65 expression levels. C8:0 and EPA significantly increased ABCA1 and p-JAK2, while they decreased NF-κBp65. Meanwhile, EPA had a significantly lower NF-κBp65 protein expression than that of C8:0 in LPS-stimulated ABCA1-KD cells. However, the effects of C8:0 on inflammation levels and JAK2/STAT3 pathway protein expression were somewhat inconsistent in mice and cell lines. The reason may be that ABCA1 was knocked out and knocked down in mice and cells, respectively. In addition, different tissues analyzed in animal and cell experiments may have different results. Our study showed that C8:0 plays a regulatory role in improving blood lipids and inflammation primarily through ABCA1, while EPA mainly inhibits inflammation through the TLR4/NF-κBp65 pathway. The specific mechanism is worth further exploration.

However, there are some limitations to this study. (1) In the mouse experiment, the sample size was small because the ABCA1 homozygote mice could not reproduce. (2) The effects of ABCA1 knockout on different inflammatory pathways are unclear, and there are compensatory mechanisms, which will affect the results of the study. (3) In the future, we should observe the effects of C8:0 on inflammation and atherosclerosis in ABCA1- and apoE-gene-deficient mice. (4) The binding protein of C8:0 is still unknown and may be the key protein for its function; therefore, further study is necessary. (5) Dyslipidemia and inflammatory responses play a key role in the progression of atherosclerosis. In our study, no atherosclerotic lesions were found in ABCA1^−/−^ mice fed a high-fat or palmitate diet for 8 weeks, which is consistent with previous findings [19,45]. Obviously, the mechanism of ABCA1 knockout in AS needs to be further studied, as well as the existing compensatory mechanism and the changing mechanism of the effects of C8:0 and EPA.

## 5. Conclusions

ABCA1 plays an important role in the regulation of lipid metabolism and inflammatory pathways. Our data showed that ABCA1 deficiency resulted in dyslipidemia and increased inflammation in mice, and that ABCA1 knockdown promoted increased inflammatory levels in RAW 264.7 cells. We found that EPA significantly improved cholesterol metabolism, while C8:0 showed only a significant decrease in TG. In addition, EPA inhibited inflammation levels significantly better than C8:0 in both ABCA1^−/−^ mice and ABCA1-KD cells. These results differ from our previous studies of C57BL/6J mice and RAW 264.7 cells. The present study suggests that C8:0 can inhibit inflammation and improve blood lipids primarily through the upregulation of ABCA1 and p-JAK2/p-STAT3, while EPA can inhibit inflammation primarily through the TLR4/NF-κBp65 signaling pathway. The upregulation of the ABCA1 expression pathway by functional nutrients may provide research targets for the prevention and treatment of AS.

## Figures and Tables

**Figure 1 nutrients-15-01296-f001:**
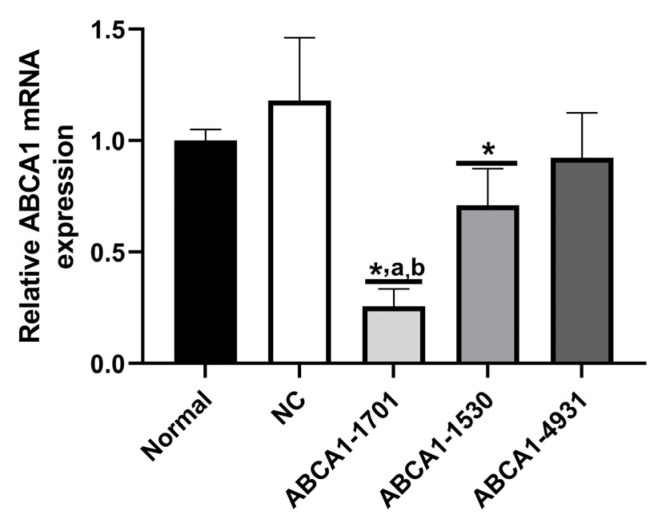
Effective plasmid screening of ABCA1 siRNA. Data in the figure are expressed as the mean ± SD with three samples in each group (n = 3). * *p* < 0.05, versus NC group; ^a^ *p* < 0.05, versus ABCA1-1530 group; ^b^ *p* < 0.05, versus ABCA1-4931 group.

**Figure 2 nutrients-15-01296-f002:**
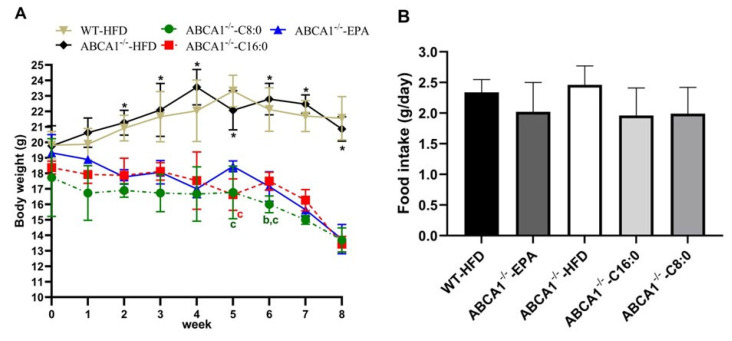
Fasting body weight and food intake of ABCA1^−/−^ mice. (**A**) The fasting body weight was determined once per week; (**B**) Average daily food intake during the experiment. The data in the figure are expressed as mean ± SD with five samples in each group (n = 5). * *p* < 0.05 versus the ABCA1^−/−^-C8:0, ABCA1^−/−^-C16:0, and ABCA1^−/−^-EPA groups; ^b^ *p* < 0.05 versus the ABCA1^−/−^-C16:0 group; ^c^ *p* < 0.05 versus the ABCA1^−/−^-EPA group.

**Figure 3 nutrients-15-01296-f003:**
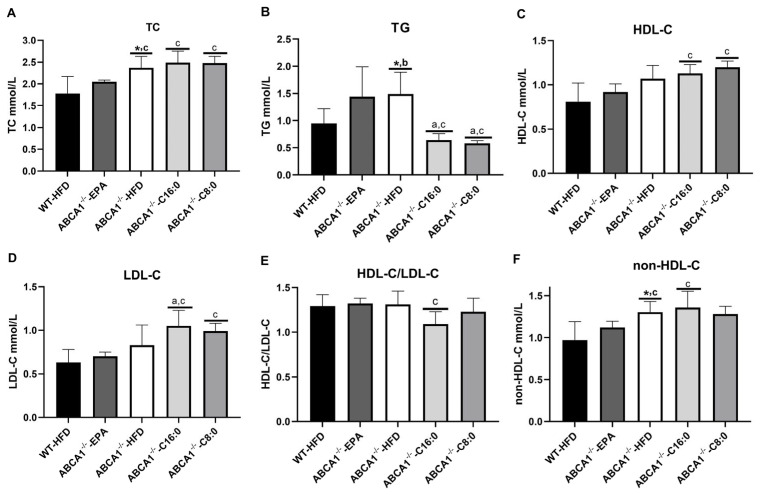
Serum lipid profiles in ABCA1^−/−^ mice. (**A**) TC, (**B**) TG, (**C**) HDL-C, (**D**) LDL-C, (**E**) HDL-C/LDL-C, and (**F**) non-HLD-C. The data in the figure are expressed as mean ± SD with five samples in each group (n = 5). * *p* < 0.05 versus the WT-HFD group; ^a^ *p* < 0.05 versus the ABCA1^−/−^ -HFD group; ^b^ *p* < 0.05 versus the ABCA1^−/−^-C16:0 group; ^c^ *p* < 0.05 versus the ABCA1^−/−^-EPA group.

**Figure 4 nutrients-15-01296-f004:**
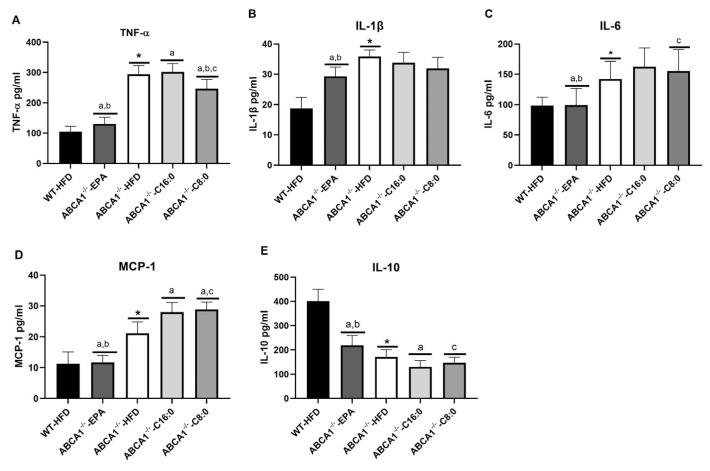
Serum levels of inflammatory cytokines in ABCA1^−/−^ mice. (**A**) TNF-α, (**B**) IL-1β, (**C**) IL-6, (**D**) MCP-1, and (**E**) IL-10. The data in the figure are expressed as mean ± SD with five samples in each group (n = 5). * *p* < 0.05 versus the WT-HFD group; ^a^ *p* < 0.05 versus the ABCA1^−/−^-HFD group; ^b^ *p* < 0.05 versus the ABCA1^−/−^-C16:0 group; ^c^ *p* < 0.05 versus the ABCA1^−/−^-EPA group.

**Figure 5 nutrients-15-01296-f005:**
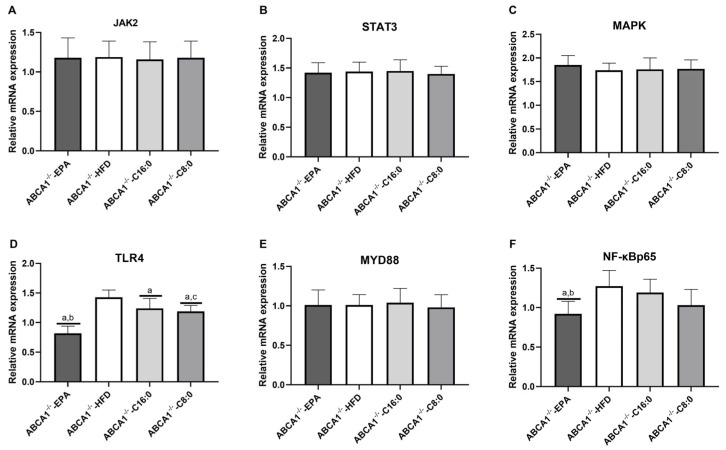
The mRNA expression levels of the signaling components TLR4 and JAK2/STAT3 in the ABCA1^−/−^ mouse aorta. (**A**) JAK2, (**B**) STAT3, (**C**) MAPK, (**D**) TLR4, (**E**) MYD88, and (**F**) NF-κBp65. The data in the figure are expressed as mean ± SD with four samples in each group (n = 4). ^a^ *p* < 0.05 versus the ABCA1^−/−^-HFD group; ^b^ *p* < 0.05 versus the ABCA1^−/−-^-C16:0 group; ^c^ *p* < 0.05 versus the ABCA1^−/−^-EPA group.

**Figure 6 nutrients-15-01296-f006:**
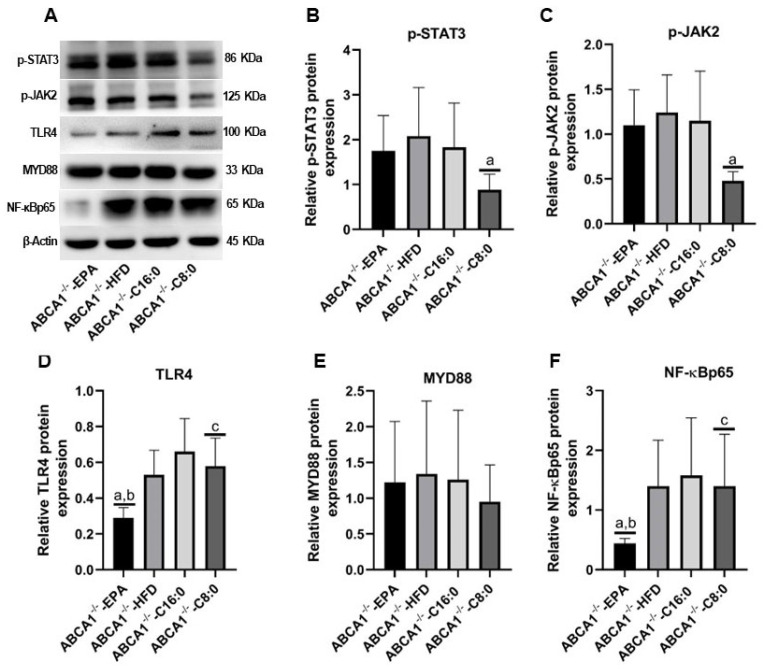
Relative protein expression of the TLR4 and JAK2/STAT3 signaling components in the ABCA1^−/−^ mouse aorta. (**A**) bolt sections, (**B**) p-STAT3, (**C**) p-JAK2, (**D**) TLR4, (**E**) MYD88, and (**F**) NF-κBp65. The data in the figure are expressed as mean ± SD with four samples in each group (n = 4). ^a^ *p* < 0.05 versus the ABCA1^−/−^-HFD group; ^b^ *p* < 0.05 versus the ABCA1^−/−^-C16:0 group; ^c^ *p* < 0.05 versus the ABCA1^−/−^-EPA group.

**Figure 7 nutrients-15-01296-f007:**
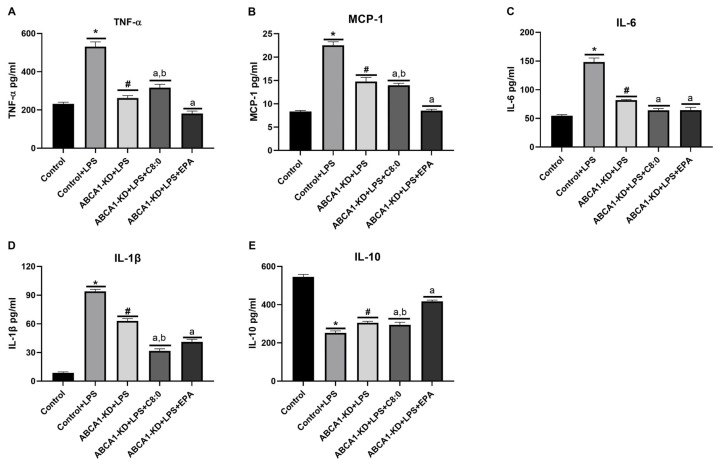
Inflammatory cytokine levels in LPS-stimulated ABCA1-KD RAW 264.7 cells. (**A**) TNF-α, (**B**) MCP-1, (**C**) IL-6, (**D**) IL-1β, and (**E**) IL-10. The data in the figure are expressed as mean ± SD with five samples in each group (n = 5). * *p* < 0.05 versus the control group; ^#^ *p* < 0.05 versus the control + LPS group; ^a^ *p* < 0.05 versus the ABCA1-KD + LPS group; ^b^ *p* < 0.05 versus the ABCA1 -KD + LPS + EPA group.

**Figure 8 nutrients-15-01296-f008:**
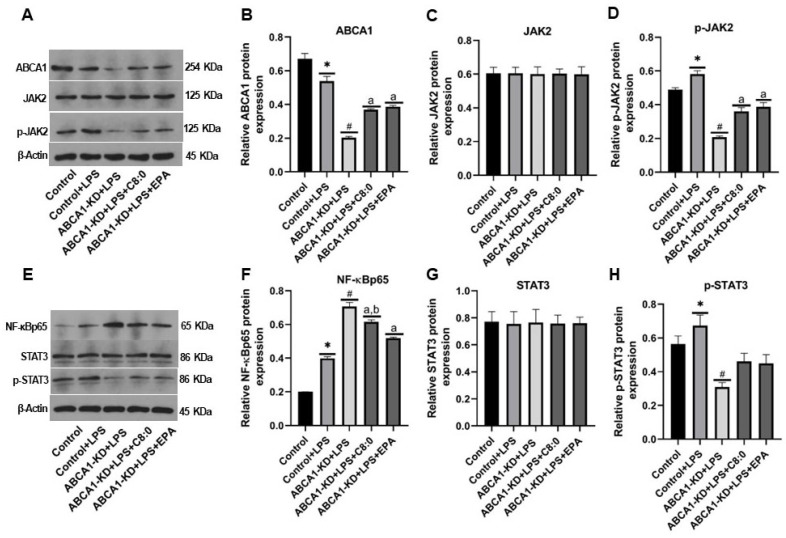
The relative protein expression of JAK2/STAT3 signaling components in LPS-stimulated ABCA1-KD RAW 264.7 cells. (**A**) and (**E**) bolt sections, (**B**) ABCA1, (**C**) JAK2, (**D**) p-JAK2, (**F**) NF-κBp65, (**G**) STAT3, and (**H**) p-STAT3. The data in the figure are expressed as mean ± SD with three samples in each group (n = 3). * *p* < 0.05 versus the control group; ^#^ *p* < 0.05 versus the control + LPS group; ^a^ *p* < 0.05 versus the ABCA1-KD + LPS group; ^b^ *p* < 0.05 versus the ABCA1-KD + LPS + EPA group.

**Table 1 nutrients-15-01296-t001:** Real-time PCR primer sequences.

Indicators	Primer	Sequence	Primer Bank ID
β-actin	Forward	5′-GGCTGTATTCCCCTCCATCG -3′	6671509a1
	Reverse	5′-CCAGTTGGTAACAATGCCATGT -3′	
JAK2	Forward	5′-TTGTGGTATTACGCCTGTGTATC-3′	6680508a1
	Reverse	5′-ATGCCTGGTTGACTCGTCTAT-3′	
STAT3	Forward	5′-CAATACCATTGACCTGCCGAT-3′	13277852a1
	Reverse	5′-GAGCGACTCAAACTGCCCT-3′	
MAPK	Forward	5′-GGCTCGGCACACTGATGAT-3′	6754632a1
	Reverse	5′-TGGGGTTCCAACGAGTCTTAAA-3′	
TLR4	Forward	5′-ATGGCATGGCTTACACCACC-3′	10946594a1
	Reverse	5′-GAGGCCAATTTTGTCTCCACA-3′	
MYD88	Forward	5′-TCATGTTCTCCATACCCTTGGT-3′	26354939a1
	Reverse	5′-AAACTGCGAGTGGGGTCAG-3′	
NF-κBp65	Forward	5′-CACCGGATTGAAGAGAAGCG-3′	30047197a1
	Reverse	5′-AAGTTGATGGTGCTGAGGGA-3′	

JAK2, Janus kinase 2; STAT3, signal transducer and activator of transcription 3; MAPK, mitogen-activated protein kinase; TLR4, toll-like receptor 4; MYD88, myeloid differentiation primary response 88; NF-κBp65, nuclear factor kappa Bp65.

## Data Availability

Not applicable.

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
