# Peer review of "Effects of Caprylic Acid and Eicosapentaenoic Acid on Lipids, Inflammatory Levels, and the JAK2/STAT3 Pathway in ABCA1-Deficient Mice and ABCA1 Knock-Down RAW264.7 Cells"

_nutrients, 2023, doi:10.3390/nu15051296_

Round 1
Reviewer 1 Report
The manuscript by Zhang et al. describes a study investigating the mechanism through which caprylic acid exerts anti-inflammatory and lipid-lowering effects, using in vivo and in vitro models where the ATP-binding cassette transporter A1 gene expression was suppressed.
The authors conclude that, compared with wild-type animals, caprylic acid significantly reduces triglycerides, while it has no effect on cholesterol, while EPA omega-3 has potent lowering effect on cholesterol and inflammation. These effects would be the result of the activation of two different signalling pathways.
The experiments were conducted in a correct way, and the conclusions are supported by the results obtained. References are appropriate and the figures are of acceptable quality.
However, there are a few points unclear to me:
Why do ABCA1-/- mutants on a high-fat diet had higher HDL-cholesterol levels than wild types on the same diet? Shouldn't they be lower (line 346)? Usually, in ABCA1-/- mutants, HDL-cholesterol remains unchanged following a high-fat diet, while it may increase in ABCA1+/- heterozygotes. Are the authors sure that the animals were homozygotes -/- ?
In the discussion, it is true that reference no. 20 mentioned insufficient absorption of vitamins and impaired platelet aggregation but I cannot find any mention of decreased survival of the animals.
The discussion could be further shortened, especially where it is a mere repetition of the results of the experiments (lines 376-380, and the conclusions).
Minor
Line 29. In the abstract and in sixteen occurrences of the text, the macrophage cell line is sometimes referred to as RAW246.7. The correct term is instead RAW 264.7
Line 34. If possible, avoid repeating the word “showed”
Lines 50-51. If possible, change the verb “believe”
Line 57. Better: cholesterol efflux (see also line 77)
Line 106. Homozygote is a noun, while homozygous is an adjective. I think “ABCA1 homozygotes” is better
Line 106. The period before “Although” should be deleted
Lines 176-178. The sentence seems to be taken from an instruction manual, if possible rewrite.
Line 179. The reference should be written in brackets
Line 192. Please correct “derivation” into “deviation”
Line 207. In figure 1 correct HDF as HFD, and the same in the remaining 8 occurrences throughout the text. Also, if possible, indicate the mutants as ABCA1-/- and not ABCA-/-
Lines 234-235. The lower level of TNF-alpha refers to the group of animals treated with EPA not C8:0. Please double check
Line 313. Correct ACBA1 to ABCA1
Line 315. When citing reference 19, remember that Robert is the name of the first author while his surname is Aiello. Therefore, the sentence must start as “Aiello et al. reported…”
Author Response
Response to Reviewer 1 Comments
The manuscript by Zhang et al. describes a study investigating the mechanism through which caprylic acid exerts anti-inflammatory and lipid-lowering effects, using in vivo and in vitro models where the ATP-binding cassette transporter A1 gene expression was suppressed.
The authors conclude that, compared with wild-type animals, caprylic acid significantly reduces triglycerides, while it has no effect on cholesterol, while EPA omega-3 has potent lowering effect on cholesterol and inflammation. These effects would be the result of the activation of two different signalling pathways.
The experiments were conducted in a correct way, and the conclusions are supported by the results obtained. References are appropriate and the figures are of acceptable quality.
However, there are a few points unclear to me:
Point 1: Why do ABCA1-/- mutants on a high-fat diet had higher HDL-cholesterol levels than wild types on the same diet? Shouldn't they be lower (line 346)? Usually, in ABCA1-/- mutants, HDL-cholesterol remains unchanged following a high-fat diet, while it may increase in ABCA1+/- heterozygotes. Are the authors sure that the animals were homozygotes -/- ?
Response 1: Thank you for your detailed and careful analysis. In fact, the present study found that ABCA1 knockout mice had increased HDL-C levels compared to WT mice fed a high-fat diet, although there were no significant differences, which is inconsistent with previous findings (references 28). ABCA1 deficiency leads to lower HDL levels, as in Tangier's disease, resulting in high TG and TC and low HDL levels. Laura et al (Calpe-Berdiel L, Rotllan N, Palomer X, et al. Biochim Biophys Acta. 2005,1738(1-3):6-9. PMID: 16413225). used a chow diet to feed ABCA1+/+, ABCA1+/- and ABCA1-/- mice for three months and detected plasma HDL-C (mM) of 1.1±0.3, 0.6±0.1, and 0.03±0.04, but plasma TC (mM) was 2.0±0.6, 1.3±0.2, and 0.5±0.1, respectively, and the TC levels were not consistent with previous studies (references 28). Drobnik et al. used cholesterol-free diets to feed ABCA1+/+ and ABCA1-/- mice for 14 days and detected a significant reduction in serum TC in ABCA1-/- mice as well (references 29). On the contrary, in the present study, TG and TC levels were significantly higher in ABCA1-/- mice fed a high-fat diet for 8 weeks, suggesting that different diets and the duration of intervention may also affect differences in lipid metabolism. Haghpassand et al. reported that high fat feeding increased HDL cholesterol and apoAI levels in wild-type mice or bone marrow transplanted ABCA-/- mice (references 30). Additionally, no AS formation was found in the arteries of ABCA1-/- mice in this study, which is consistent with previous studies (references 19). This also suggests that there is a compensatory mechanism for lipid metabolism in mice after ABCA1 knockout. A single deficiency of ABCA1 or ABCG1 in macrophages has been reported to not increase atherosclerosis, probably because ABCA1 deficiency leads to up-regulation of ABCG1 expression (references 31). Similarly, ABCG1 deficient mice were shown to have decreased plasma HDL cholesterol levels when fed a high cholesterol diet. Furthermore, liver SR-BI plays a pivotal role in HDL cholesterol clearance from plasma and, consequently, plasma HDL cholesterol levels. ABCA1 knockdown may cause up-regulation of these proteins while compensating for dyslipidaemia, which deserves further investigation. Thank you for your reminder, we have added a note to the discussion accordingly (lines 347-354).
We determined that the mice used ABCA1 knockout mice and identified the genotype by polymerase chain reaction (PCR). Furthermore, we often use the HDL-C/LDL-C ratio to assess the HDL level, and in this study we found no significant change in the HDL-C/LDL-C ratio in the two groups of mice. Additionally, both TG and TC levels affect HDL-C levels, and LDL-C and non-HDL-C levels are often used clinically as the main criteria for lipid-modifying therapy. We added the non-HDL-C index (non-HDL-C = TC-HDL-C) and found that non-HDL-C was significantly higher in ABCA1 knockout mice compared to WT mice, while the EPA group had a significantly lower level of non-HDL-C than mice in the C8:0 and C16:0 groups (Figure 3).
Point 2: In the discussion, it is true that reference no. 20 mentioned insufficient absorption of vitamins and impaired platelet aggregation but I cannot find any mention of decreased survival of the animals.
Response 2: Thank you for your reminder. Homozygous female ABCA1 deficient mice are difficult to breed, have a marked reduction in the number of pregnancies and produce small litter sizes, most likely due to altered hormone production caused by reduced estrogen and progesterone levels and subsequent placental abnormalities (references 21); and ABCA1 knockout may cause insufficient vitamin absorption and platelet aggregation, as well as severe small intestinal lesions, which also contribute to the reduced survival rate (references 20). Furthermore, homozygous ABCA1 deficient pups produced by mating of ABCA1 deficient males and females were found to be less likely to survive to weaning at 3 weeks of age. Therefore, we used male C57BL/6J mice and heterozygous ABCA1 mice to reproduce more homozygous mice. This point was not clearly stated by us and was added in the corresponding discussion (lines 317-320).
Point 3: The discussion could be further shortened, especially where it is a mere repetition of the results of the experiments (lines 376-380, and the conclusions).
Response 3: We gratefully acknowledge your suggestions. We have revised the presentation of experimental results in the discussion and condensed the conclusions ( lines 380-383, lines 400-404 and the conclusions).
Minor
Point 4: Line 29. In the abstract and in sixteen occurrences of the text, the macrophage cell line is sometimes referred to as RAW246.7. The correct term is instead RAW 264.7
Response 4: We thank you for pointing this out. We have revised it.
Point 4: Line 34. If possible, avoid repeating the word “showed”
Response 4: Thank you for your advice. We have changed “Our study showed that EPA showed better effects than C8:0……” to “Our study showed that the EPA had better effects than C8:0……”(line 34).
Point 5: Lines 50-51. If possible, change the verb “believe”
Response 5: Thank you for your advice. We have changed “Current studies believe that ABCA1 can not only promote……” to “Current studies have shown that ABCA1 not only promotes……”.(line 50)
Point 6: Line 57. Better: cholesterol efflux (see also line 77)
Response 6: We agree and have updated.
Point 7: Line 106. Homozygote is a noun, while homozygous is an adjective. I think “ABCA1 homozygotes” is better
Response 7: Thank you for your advice.We have fixed the error.
Point 8: Line 106. The period before “Although” should be deleted
Response 8: Thank you for your advice. There is no "although" in this paragraph, so I understand that the period before "We" should be replaced by a comma (line 106).
Point 9: Lines 176-178. The sentence seems to be taken from an instruction manual, if possible rewrite.
Response 9: Thank you for your advice. We've described it briefly. The new sentence as follows:” The 20 mg of mouse aorta tissue sample was extracted by protein lysis buffer and the cells were extracted with RIPA buffer (CST)” (lines 179-180).
Point 10: Line 179. The reference should be written in brackets
Response 10: We thank you for pointing this out. We have revised it.
Point 11: Line 192. Please correct “derivation” into “deviation”
Response 11: Thank you for your advice. We have revised it.
Point 12: Line 207. In figure 1 correct HDF as HFD, and the same in the remaining 8 occurrences throughout the text. Also, if possible, indicate the mutants as ABCA1-/- and not ABCA-/-
Response 12: We thank you for pointing this out. We have revised HDF as HFD in figure 1 and throughout the text. Also, revised ABCA-/- as ABCA1-/-.
Point 13: Lines 234-235. The lower level of TNF-alpha refers to the group of animals treated with EPA not C8:0. Please double check
Response 13: Thank you for your detailed and careful analysis. This refers to the C8:0 group, the latter is written with a duplicate C8:0, which should be the HFD group, and we have corrected.
Point 14: Line 313. Correct ACBA1 to ABCA1
Response 14: We thank you for pointing this out. We have revised it.
Point 15: Line 315. When citing reference 19, remember that Robert is the name of the first author while his surname is Aiello. Therefore, the sentence must start as “Aiello et al. reported…”
Response 14: Thank you for your advice. We have checked the full text and revised them.
We would like to thank the referee again for taking the time to review our manuscript. Please see the attachment for the revised manuscript.

Reviewer 2 Report
The manuscript reports on the effects of caprylic acid, palmitic acid and eicosapentaenoic acid on lipid profile, inflammatory markers and JAK2/STAT3 pathway in ABCA1-deficient mice and ABCA1 knock-down RAW264.7 cells. The expected effects were obtained by using C8:0 and EPA acids.
The manuscript is interesting, the results presented in it are of great cognitive importance, however, the way they are presented needs definite improvement.
Minor remarks
My first comment concerns the title of the manuscript. I think it should also include a reference to EPA in it (e.g. Effects of Caprylic Acid and Eicosapentaenoic Acid on...).
Section 2.3 states that the high-fat diet was replaced with an intervention diet for 1 week. Whereas two sentences later the statement "During the 8-week intervention..." follows. These two sentences are contradictory and should be corrected.
The sentence "Because ABCA1 homozygous cannot reproduce and has a 50% mortality rate." It is unfinished. Its logical completion is the next sentence, which should begin after a comma, not a period.
Point 2.5 - in my opinion, Figure S1 should be in the text of the manuscript.
Point 2.9 - instead of "previous studies18" should be previous studies [18].
Figures 1-7 - the passage "C8:0, caprylic acid; C16:0, palmitic acid; EPA, eicosapentaenoic acid; HFD, high-fat diet." should be removed from the descriptions under the figures.
Throughout the manuscript, the abbreviations HFD and HDF are used interchangeably; it would have been better if the Authors had opted for one or the other. A similar comment applies to the notation ABCA-/- (in the text) or ABCA1-/- (in the figures).
Point 3, namely Results needs rewriting. The descriptions about the different Figures are written chaotically and are sometimes not very understandable (too much different information in one sentence). It would also be better if the Authors described the subsequent Figures one by one, rather than in a haphazard manner.
A misquotation appears in line 315. Instead of "Robert et al [19]" it should be Aiello et al [19].
Author Response
Response to Reviewer 2 Comments
The manuscript reports on the effects of caprylic acid, palmitic acid and eicosapentaenoic acid on lipid profile, inflammatory markers and JAK2/STAT3 pathway in ABCA1-deficient mice and ABCA1 knock-down RAW264.7 cells. The expected effects were obtained by using C8:0 and EPA acids.
The manuscript is interesting, the results presented in it are of great cognitive importance, however, the way they are presented needs definite improvement.
Minor remarks
Point 1: My first comment concerns the title of the manuscript. I think it should also include a reference to EPA in it (e.g. Effects of Caprylic Acid and Eicosapentaenoic Acid on...).
Response 1: Thank you for your valuable suggestions. The title of the article has been changed to: Effects of Caprylic Acid and Eicosapentaenoic Acid on Lipids, Inflammatory Levels, and the JAK2/STAT3 Pathway in ABCA1-deficient mice and ABCA1 knock-down RAW264.7 cells.
Point 2: Section 2.3 states that the high-fat diet was replaced with an intervention diet for 1 week. Whereas two sentences later the statement "During the 8-week intervention..." follows. These two sentences are contradictory and should be corrected.
Response 2: We thank you for pointing this out. We have revised the states. The new sentence as follows:” Both the ABCA1-/- mice and the same week old C57BL/6J mice were kept on a HFD prior to intervention, and the HFD was replaced with intervention diets for 8 weeks after random allocation based on fasting weight……” (line 112-114).
Point 3: The sentence "Because ABCA1 homozygous cannot reproduce and has a 50% mortality rate." It is unfinished. Its logical completion is the next sentence, which should begin after a comma, not a period.
Response 3: We thank you for pointing this out. We have revised it.
Point 4: 2.5 - in my opinion, Figure S1 should be in the text of the manuscript.
Response 4: Thank you for your advice. We have shown Figure S1 as Figure1 in the text of the manuscript.
Point 5: 2.9 - instead of "previous studies18" should be previous studies [18].
Response 5: We thank you for pointing this out. We have revised it.
Point 6: Figures 1-7 - the passage "C8:0, caprylic acid; C16:0, palmitic acid; EPA, eicosapentaenoic acid; HFD, high-fat diet." should be removed from the descriptions under the figures.
Response 6: Thank you for your advice. We have removed the passage "C8:0, caprylic acid; C16:0, palmitic acid; EPA, eicosapentaenoic acid; HFD, high-fat diet" from the descriptions under the figures 1-8 of the revised manuscript.
Point 7:Throughout the manuscript, the abbreviations HFD and HDF are used interchangeably; it would have been better if the Authors had opted for one or the other. A similar comment applies to the notation ABCA-/- (in the text) or ABCA1-/- (in the figures).
Response 7: We thank you for pointing this out. We have revised HDF as HFD in figure 1 and throughout the text. Also, revised ABCA-/- as ABCA1-/-.
Point 8:namely Results needs rewriting. The descriptions about the different Figures are written chaotically and are sometimes not very understandable (too much different information in one sentence). It would also be better if the Authors described the subsequent Figures one by one, rather than in a haphazard manner.
Response 8: We gratefully acknowledge your suggestions. We have revised the statement of result to address your concerns and hope that it is now clearer. Please see the result part of the revised manuscript, lines 214-221, lines 232–237, lines 275-282 and lines 295-300.
Point 9:A misquotation appears in line 315. Instead of "Robert et al [19]" it should be Aiello et al [19].
Response 9: Thank you for your advice. We have checked the full text and revised them.
We would like to thank the referee again for taking the time to review our manuscript. Please see the attachment for the revised manuscript.

Reviewer 3 Report
In the current manuscript, the authors aim to explore the roles of caprylic acid on lipid and inflammatory levels and the involved mechanisms in mice and the mouse macrophage cell line. Below are my comments:
1. Please proofread the current manuscript and correct typos. For example, line 238, 252, 265, 266, Figure 1, "HFD" instead of "HDF".
2. In Figure 1, what is the reason of the body weight drop in almost all groups from day 5?
3. What is the reason for choosing the Raw264.7 cell line in this study? Basically, the effects of C8 on inflammatory factors and cell signaling pathways are not consistent in mice and in the cell line based on current results.
Author Response
Response to Reviewer 3 Comments
In the current manuscript, the authors aim to explore the roles of caprylic acid on lipid and inflammatory levels and the involved mechanisms in mice and the mouse macrophage cell line. Below are my comments:
Point 1:Please proofread the current manuscript and correct typos. For example, line 238, 252, 265, 266, Figure 1, "HFD" instead of "HDF".
Response 1: We thank you for pointing this out. We have revised HDF as HFD in figure 1 and throughout the text.
Point 2:In Figure 1, what is the reason of the body weight drop in almost all groups from day 5?
Response 2: Thank you for your detailed and careful analysis. After 2 weeks, the fasting body weight of ABCA knockout mice in the EPA, C8:0 and C16:0 groups began to decrease significantly compared to that of the HFD group. And all ABCA1-/- mice began to lose weight significantly at 5 weeks. We believe it is mainly due to ABCA1 deficiency. Homozygous female ABCA1 deficient mice have been reported to be difficult to breed, have a marked reduction in the number of pregnancies, and produce small litter sizes, most likely due to altered hormone production caused by reduced estrogen and progesterone levels and subsequent placental abnormalities (references 21); and ABCA1 knockout may cause insufficient vitamin absorption and platelet aggregation, as well as severe small intestinal lesions, which are also responsible for reduced survival and weight loss (references 20). Furthermore, homozygous ABCA1 deficient pups produced by mating of ABCA1 deficient males and females were found to be less likely to survive to weaning at 3 weeks of age. Therefore, we used male C57BL/6J mice and heterozygous ABCA1 mice to reproduce more homozygous mice. Although there were no significant differences in the average feed intake among all groups during the intervention period, the feed intake of the EPA, C8:0, and C16:0 groups was relatively low, so the body weight of the mice in the three groups remained generally low. Our previous studies have confirmed that EPA and C8:0 can reduce fasting body weight in C57BL/6J mice (references 15) , and combined with inflammation in ABCA1-/- mice, was more likely to lead to weight loss. Of course, it may be that ABCA1 knockout mice lost weight for some unknown reason. Thank you for your reminder; we have added a note to the discussion accordingly (lines 317-320).
Point 3:What is the reason for choosing the Raw264.7 cell line in this study? Basically, the effects of C8 on inflammatory factors and cell signaling pathways are not consistent in mice and in the cell line based on current results.
Response 3: We are thankful for your professional review work on our article.
(1) RAW264.7 is a mouse peritoneal macrophage cell line and one of the most commonly used inflammatory cell models. Furthermore, the phagocytic cholesterol of RAW 264.7 cells, which forms foam cells, plays an important role in the formation of atherosclerosis. The ABCA1 transporter plays an important role in the accumulation of lipids in macrophages and the release of inflammatory factors. Targeting ABCA1 in macrophages may help reduce foam cell formation, relieve the vascular inflammatory environment, and inhibit the development of atherosclerosis. ABCA1 is broadly expressed with high levels in macrophages, liver cells, intestinal cells, adrenal gland, endothelial cells, and placental trophoblast (references 19). Our previous experiments have confirmed that MCFAs can upregulate the expression of the ABCA1 gene and protein in the apoE-deficient mouse liver, and C8:0 can promote cholesterol efflux in macrophages (references 14). Further studies found that C8:0 could inhibit the levels of inflammatory cytokines in RAW 264.7 cells and upregulate ABCA1 expression in the mouse aorta and in RAW264.7, and confirmed that C8:0 could inhibit the inflammatory response based on the ABCA1-mediated JAK2/STAT3 pathway (references 15). RAW264.7 cells are involved in the regulation of lipid metabolism and inflammatory response, and are key cells in the formation of atherosclerosis. In addition, RAW264.7 cells are mostly used in the previous research bases and references, so we chose the RAW264.7 cell line in this study. Of course, more studies on the role of ABCA1 in other cells are worth studying.
(2) In mouse experiments, homozygous ABCA1 knockout mice were obtained by breeding heterozygous ABCA1 mice, and were confirmed as ABCA1-/- mice through genetic identification. while, in cell experiments, we screened out the most effective plasmid for inhibiting ABCA1 (Figure 1 in the revised manuscript), ABCA1-1701 had the strongest inhibitory level, and its expression decreased by 74.3% compared with normal cells, and the ABCA1-KD RAW 264.7 cells were constructed with ABCA1-1701 as siRNA plasmid. In other words, mouse and cell ABCA1 was knocked out and knocked down, respectively. This may be one of the reasons why the effects of C8:0 on inflammatory cytokines and cell signaling pathways are not consistent in mice and cell lines. In addition, aorta and serum were analyzed in the mouse experiment, while RAW264.7 macrophage lysate was analyzed in the cell experiment. Different tissue cells may also have differences in performance. We will add the corresponding explanation to the discussion (lines 405-408).
We would like to thank the referee again for taking the time to review our manuscript. Please see the attachment for the revised manuscript.
